# Improving Multi-Center Generalizability of GAN-Based Fat Suppression using Federated Learning

**Pranav Kulkarni**                                    PKULKARNI@SOM.UMARYLAND.EDU
**Adway Kanhere**                                        AKANHERE@SOM.UMARYLAND.EDU
**Harshita Kukreja**                                      HKUKREJA@SOM.UMARYLAND.EDU
**Vivian Zhang**                                            VZHANG@SOM.UMARYLAND.EDU
**Paul H. Yi**                                                  PYI@SOM.UMARYLAND.EDU
**Vishwa S. Parekh**                                        VPAREKH@SOM.UMARYLAND.EDU
*University of Maryland Medical Intelligent Imaging (UM2ii) Center, Baltimore, MD 21201*

**Editors:** Accepted for publication at MIDL 2024

## Abstract

Generative Adversarial Network (GAN)-based synthesis of fat suppressed (FS) MRIs from non-FS proton density sequences has the potential to accelerate acquisition of knee MRIs. However, GANs trained on single-site data have poor generalizability to external data. We show that federated learning can improve multi-center generalizability of GANs for synthesizing FS MRIs, while facilitating privacy-preserving multi-institutional collaborations.
**Keywords:** Image Synthesis, GAN, Fat Suppression, Federated Learning

## 1. Introduction

Generative Adversarial Network (GAN)-based MRI synthesis has the potential to accelerate acquisition (Dar et al., 2019; Nie et al., 2017). One such use-case is for knee MRIs, where proton density-weighted (PD) and fluid-sensitive, fat suppressed (FS) sequences are used to detect abnormalities (Lee et al., 2011; Shakoor et al., 2018). Prior work has shown that GANs can synthesize FS sequences from non-FS PD sequences, thereby reducing acquisition times (Fayad et al., 2021). Despite exhibiting high performance, GANs trained on single-site data have poor generalizability when tested on external data due to domain shift (Dar et al., 2019; Wei et al., 2019). While curating a large, diverse, and multi-center dataset at a single site can alleviate this, it is impractical due to patient privacy. Federated Learning (FL) is a promising paradigm to facilitate multi-center collaborations to collectively train a global model without sharing patient data (Sheller et al., 2020; Dalmaz et al., 2024). In this preliminary work, we hypothesize that FL can improve multi-center generalizability of GAN-based synthesis of FS MRIs from non-FS PD knee MRIs in a privacy-preserving way.

## 2. Methods

**Datasets:** 1) An internal University of Maryland (UMB) dataset containing $n = 151$ studies with non-FS PD and FS sequences in axial and coronal planes as part of a study acknowledged as non-human subjects research by our IRB. 2) The FastMRI dataset containing $n = 7,171$ studies with non-FS PD and FS sequences in sagittal and coronal planes (Knoll et al., 2020; Zbontar et al., 2018). We randomly sampled sequence pairs for training

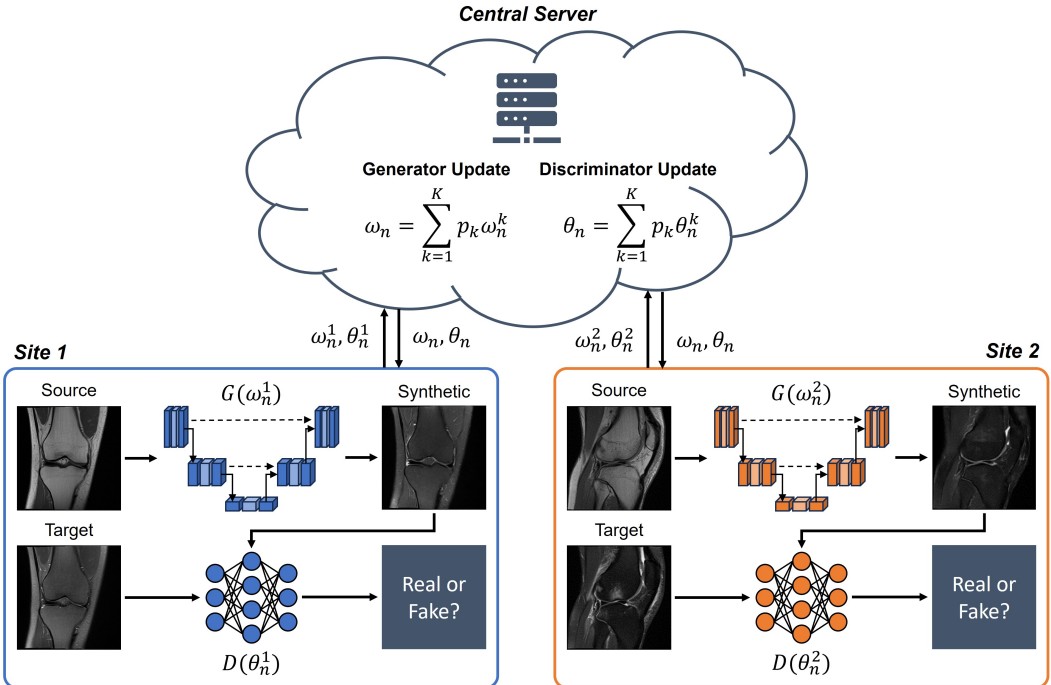

Figure 1: Privacy-preserving multi-center GAN-based synthesis of FS sequences using FL.

($n = 80$) and testing ($n = 20$). Sequence pairs were registered using ANTsPy (non-FS PD, fixed; FS, moving). Slices in the imaging plane were extracted and normalized to 0–1.

**MRI Synthesis:** We use pix2pix, a conditional GAN comprised of U-Net generator and 3-layer 70x70 PatchGAN discriminator, to synthesize FS sequences (target) from non-FS PD sequences (source) (Isola et al., 2017). The models were trained at 256x256 resolution for 200 epochs with initial LR of 5e-4 with linear decay and batch size of 1.

**Experiments:** We trained four models: 1) A single-site model with UMB data ('Baseline-UMB'). 2) A single-site model with FastMRI data ('Baseline-FastMRI'). 3) A centrally aggregated model with UMB and FastMRI data combined at a single site ('Central'). 4) A 2-client FL model with distributed UMB and FastMRI data. At the end of each epoch, client weights are communicated to the central server, aggregated using FedGAN (Rasouli et al., 2020), and communicated back to the clients (Figure 1). We compared the mean SSIM $\pm$ SD between ground-truth and synthetic FS sequences across all four models for both test sets using Wilcoxon signed-rank tests. Statistical significance was defined as $p < 0.05$.

## 3. Results

For the UMB test set, we observe that FL measures mean SSIM of $0.63 \pm 0.13$, which is comparable to Baseline-UMB ($0.64 \pm 0.13$, $p = 0.63$) and Central ($0.64 \pm 0.13$, $p = 0.74$), but significantly higher than Baseline-FastMRI ($0.46 \pm 0.11$, $p < 0.001$). For the FastMRI test set, we observe that FL measures mean SSIM of $0.58 \pm 0.12$, which is comparable to Baseline-FastMRI ($0.58 \pm 0.12$, $p = 0.99$) and Central ($0.58 \pm 0.12$, $p = 0.93$), but signifcantly higher than Baseline-UMB ($0.46 \pm 0.11$, $p < 0.001$). Examples are shown in Figure 2.

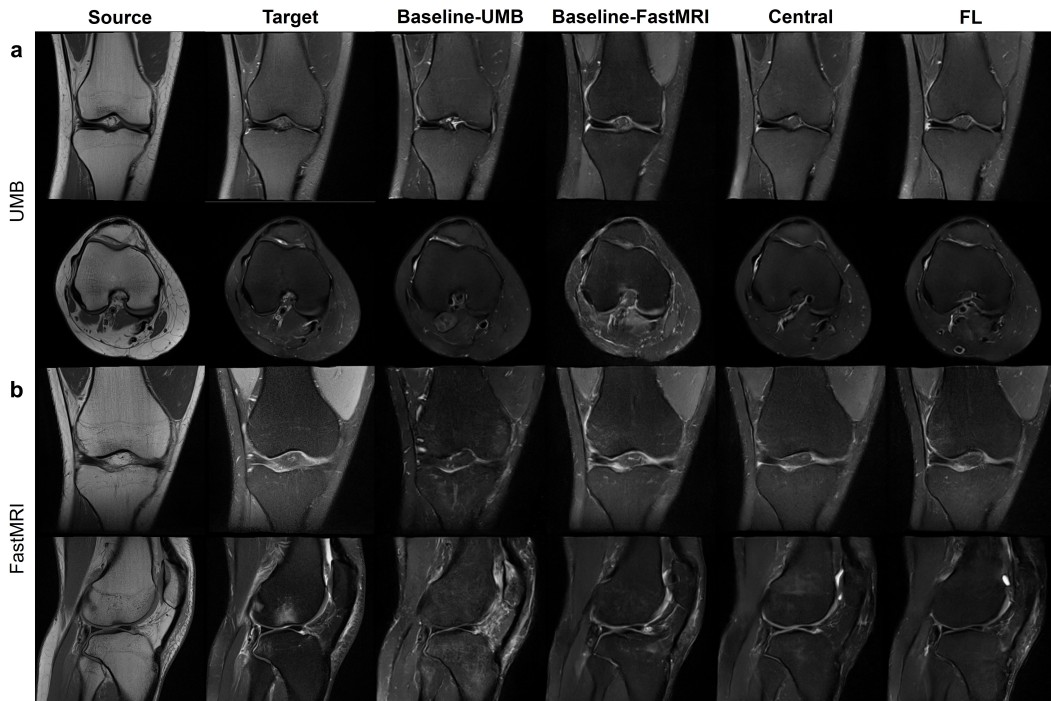

Figure 2: Examples from **(a)** UMB and **(b)** FastMRI test sets of ground-truth (cols. 1-2) with their corresponding synthetic FS sequences from the four models (cols. 3-6).

## 4. Discussion

Our results indicated two findings: 1) Single-site models had poor generalizability to external data despite exhibiting higher performance on local data. This emphasizes the importance of training GANs with larger multi-institutional datasets – a finding that aligns with prior literture (Dalmaz et al., 2024; Dar et al., 2019; Wei et al., 2019). 2) FL models exhibited significantly higher performance on external data compared to the single-site models despite the data heterogeneity between both datasets (e.g., scanner type, imaging plane).

Since our work is preliminary, it has certain limitations: 1) Our synthetic MRIs have poor mean SSIM scores. Since the GANs were trained on a small subset of both datasets, our models resulted in sub-optimal performance and can be alleviated by training on larger datasets. 2) We only use the FedGAN strategy for aggregating weights in FL. Recent literature has explored new strategies for FL with GANs (Wang et al., 2023; Dalmaz et al., 2024). For future work, we intend to address these limitations.

In conclusion, our preliminary results suggest that FL can improve the generalizability of GANs for synthesizing FS knee MRIs in the real-world while preserving patient privacy. This represents an exciting step towards synthetic MRIs becoming a clinical reality.

## Acknowledgments

This work was supported by the UMMC/UMB Innovation Challenge Award, 2023.

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
