# OpenReview forum: "Improving Multi-Center Generalizability of GAN-Based Fat Suppression using Federated Learning"
_MIDL.io/2024/Short_Papers — MIDL 2024 Short Papers_

### Official Review · Reviewer_QqFV · 2024-04-25

**Confidence:** 4
**Final Rating:** 4

**Review:**

This paper presents the application of federated learning (FL) for learning a generalizable model to synthesize fat suppressed (FS) MRIs from non-FS PD sequences for knee MRIs.

Pros:
- Results showed the FL approach performed similarly to same-site and central models while outperforming the different-site model for each of the datasets
- Statistical comparisons were presented for the experimental results
- Paper is clearly written
- The paper addresses limitations of the current study

Cons:
- Technical innovation is limited in that the paper presents an straight application of FedGAN to FS MRI synthesis problem
- Unclear why when so much data was available only a very small subset was utilized from FastMRI dataset
- The synthesized images, even for the baseline in domain or central models, appears not similar enough to the true images (as acknowledged by the authors).

---

### Decision · Program_Chairs · 2024-04-26

Accept